# Parameter Measurement of Edible Sunflower Exudates and Calibration of Discrete Element Simulation Parameters

Xiaoxiao Sun [1,2], Bin Li [1,*], Yang Liu [1] and Xiaolong Gao [1,2]

[1] Institute of Mechanical Equipment, Xinjiang Academy of Agricultural Sciences, Shihezi 832000, China; sunxiaoxiao@stu.shzu.edu.cn (X.S.); maoyinghui0701@shzu.edu.cn (Y.L.); gaoxiaolong@shzu.edu.cn (X.G.)

[2] School of Mechanical and Electrical Engineering, Shihezi University, Shihezi 832000, China

[*] Correspondence: bin175337620@shzu.edu.cn

**Abstract:** To improve the accuracy of the parameters used in discrete element simulation tests in the process of clearing fresh sunflowers, this study took the extracts of fresh sunflower as the research object and used a combination of physical experiments and simulation experiments to calibrate the discrete element simulation parameters. First, the composition of the edible sunflower extract is determined, and then, the physical test method is used to determine the characteristics of the edible sunflower extract material. Based on the results of the physical test as a simulation basis, the Plackett–Burman test is used to screen the significance of the test parameters, and the screening results show that the shear modulus, static friction coefficient, and collision recovery coefficient of sunflower kernels have significant effects on the simulated angle of repose. On the basis of the steepest climbing test, the Box–Behnken test is carried out to obtain the simulated angle of repose and significance. The second-order regression equation of texture parameters is optimized with the actual physical angle of repose (24.68°) as the target value to obtain the optimal parameter combination. Finally, through the two-sample $T$ test, $p > 0.05$ is obtained, indicating that the simulated angle of repose and the actual physical test angle of repose are not significantly different, and the relative error is 0.923%, which verifies the reliability of the optimal simulation parameter combination. The research results show that the calibrated simulation parameter combination can be used in discrete element simulation tests of the extraction process of sunflower extracts.

**Keywords:** edible sunflower extract; parameter calibration; angle of repose; discrete element

## 1. Introduction

Edible sunflower has a short growth period, strong adaptability, and low requirements for soil conditions, production area environment, ecological climate, etc., so it is widely planted worldwide [1–3]. The cleaning process is the core of the mechanical harvesting of sunflowers, and its performance directly affects important technical indicators such as the impurity rate and the seed loss rate of the sunflower harvest [4–6]. After threshing, the edible sunflower extracts enter the cleaning device. During the cleaning operation, the components of edible sunflower extracts inevitably have mutual contact and movement. Therefore, based on discrete element numerical simulation, the selection of the contact model and accurate parameter calibration of sunflower extracts can improve the correctness and accuracy of the cleaning process simulation and further optimize the relevant working parameters of the cleaning device [7–10].

At present, researchers are calibrating simulation parameters for discrete element models such as grain seeds, grain stems, soil particles, fertilizers, fruits, and vegetables. Zhang Rongfang et al. [11], Xu Fulong et al. [12], and Wang Liming et al. [13] measured the physical angles of repose of rice seeds, blue flax capsules, and pig manure, respectively, and established a regression model between the angle of repose and the significant parameters. Then, the parameters were optimized to obtain the optimal parameter combination.

Tian Xinliang et al. [14] and Wang Weiwei et al. [15] selected the Hertz–Mindlin with JKR contact parameter model to calibrate the discrete element simulation parameters of a corn kernel–soil mixture and corn stalk powder, respectively, taking the actual physical accumulation angle as the target. The saliency parameters are optimized to obtain the optimal parameter combination. Wang Wanzhang et al. [16] used a multiscale particle aggregation method to establish a discrete element simulation model of a wheat plant and calibrated the parameters in combination with a bench test and a simulation test. BOAC et al. [17] took soybeans as the research object and used the physical parameters and contact model parameters of existing soybeans as the basis for the selection of discrete element simulation parameters. They compared the simulation angles of repose of soybeans under the single-sphere particle model and the multisphere particle model. The single-sphere particle model can be better simulated, and the optimal parameter combination can be obtained. In summary, most domestic and foreign research scholars have mainly calibrated discrete element simulation parameters for soil, corn, and wheat stalks, rarely used more comprehensive physical tests to determine the material parameters of sunflower extract, and calibrated the discrete element model simulation parameters of sunflower extract based on the actual parameters.

In this paper, the measured basic physical and mechanical parameters are used as the basis for the selection of discrete element simulation parameters, and the actual measured physical angle of repose is used as the target value. The Plackett–Burman test, the steepest climbing test and the Box–Behnken test are carried out in sequence. The discrete element simulation parameters of the exudate from sunflower are calibrated. Double-sample T detection is performed on the simulated angle of repose and the actual physical angle of repose to prove the authenticity of the simulation test and finally obtain the optimal parameter combination. This research can provide a reliable discrete element simulation parameter basis for the movement law of sunflower in the cleaning process and optimization of the cleaning device.

## 2. Materials and Methods

The object of the study in this article is edible sunflower extracts from the planting area of Urumqi County, Xinjiang. The sampled variety of edible sunflower is Sanrui No. 39. Sunflower extract is a typical bulk material. The basic physical and mechanical characteristics include the composition, shape and size, moisture content, density, elastic modulus, shear modulus, Poisson's ratio, and static friction coefficient of the sunflower extract and the collision recovery factor.

### 2.1. Composition of the Exudate

One thousand grams of the material was randomly sampled each time, and then the sample was classified. It can be concluded that the edible sunflower extract contains edible sunflower seeds, edible sunflower plates (broken), edible sunflower petioles, and light impurities. The utilization accuracy was 0.01 g. An electronic scale was used to weigh the four components separately; sampling was repeated 3 times and the average value was taken to obtain the mass percentage of the sunflower seeds, sunflower plates (broken), sunflower petioles, and light impurities in the sunflower extract, as shown in Figure 1. Since light impurities are directly blown out under the action of wind during the cleaning process, they have almost no effect on the cleaning performance, so only sunflower seeds, sunflower disks (broken), and sunflower petioles were used for the material property analysis.

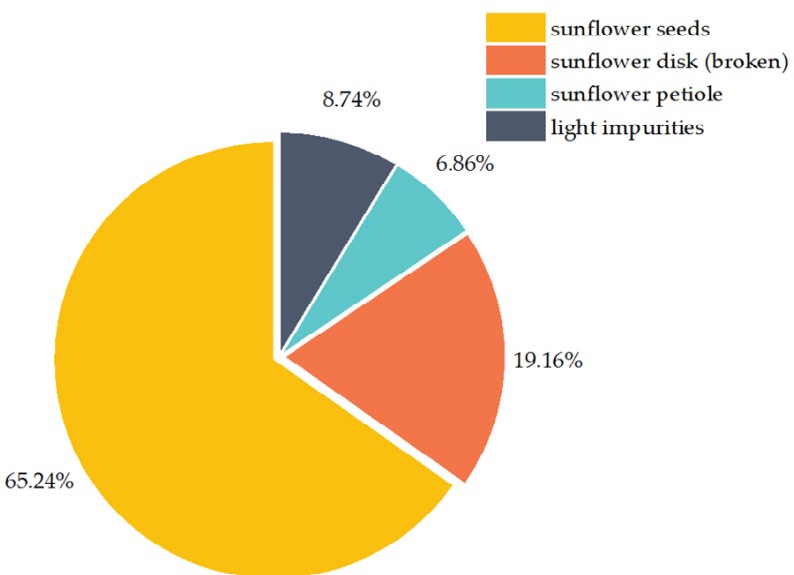

**Figure 1.** Percentage of edible sunflower extracts.

*2.2. Determination of Physical Characteristic Parameters*

We randomly selected 100 pieces of edible sunflower seeds, edible sunflower disks (broken), and edible sunflower petioles from the extrudates and measured their dimensional characteristics (length, diameter, and width) with an electronic digital caliper with an accuracy of 0.01 mm, as shown in Figure 2. The drying box method [18] was used to determine the moisture content of each component of edible sunflower. Thirty grams of edible sunflower seeds was sampled, and 60 g of edible sunflower disks (broken) and 60 g of edible sunflower petioles were sampled. The test was carried out 3 times in total using an electric drum. The sample was dried in an air drying oven and cooled to room temperature. The night soaking method [19] was used to measure the density of the three kinds of materials: sunflower seeds, sunflower discs (broken), and sunflower petioles.

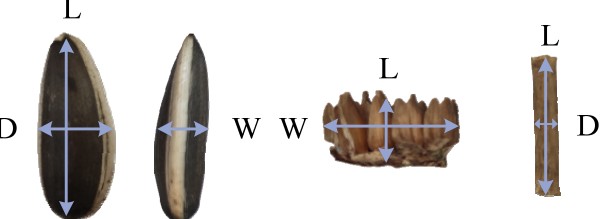

**Figure 2.** Dimension drawing of the characteristic dimensions of the extract of sunflower.

The geometric size, moisture content and average density of each component of the sunflower extract are shown in Table 1.

**Table 1.** Physical characteristics of the extracts of sunflower.

| Type | Geometry (mm) | Moisture Content (%) | Density (kg/m³) |
|---|---|---|---|
| Sunflower seeds | 27.92 × 11.18 × 6.11 (L × D × W) | 28.7 | 346 |
| Sunflower plates (broken) | 27.53 × 15.02 (L × W) | 75.75 | 679 |
| Sunflower petioles | 57.56 × 5.35 (L × D) | 78.28 | 331 |

### 2.3. Determination of Mechanical Characteristic Parameters

### 2.3.1. Elastic Modulus

The elastic modulus is an important physical quantity to measure the elasticity of materials [20]. The elastic modulus of edible sunflower exudates was determined by a compression test. Before the test, the original thickness ($T_1$) of the edible sunflower exudates was measured, and the tested sunflower exudates were naturally placed on the center of the square platform of a TA-XT plus texture analyzer. After positioning, a P/36R compression probe was used. The speed before compression was 1 mm/s, the speed during compression was 0.05 mm/s, and the speed after compression was 5 mm/s, as shown in Figure 3. The three test samples of edible sunflower seeds, edible sunflower plates (broken), and edible sunflower petioles were tested 10 times, and the average value was taken. The elastic modulus of edible sunflower seeds was calculated by Formula (1) to be 147.6 MPa. The elastic modulus of broken plates was 92.4 MPa, and the elastic modulus of the petioles of edible sunflower was 133.4 MPa.

$$E = \frac{F/A}{\Delta L/T_1} \tag{1}$$

where $E$ is the modulus of elasticity, MPa; $F$ is the applied external force, N; $A$ is the contact area, mm$^2$; $\Delta L$ is the deformation distance under the action of external force, mm; and $T_1$ is the original thickness, mm.

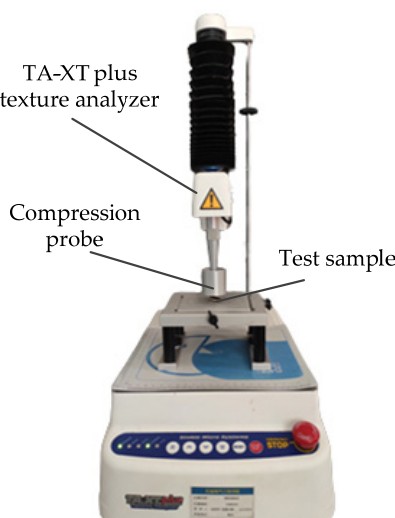

**Figure 3.** Compression test of edible sunflower extract.

### 2.3.2. Shear Modulus

The shear modulus is a material constant that refers to the ratio of shear stress to strain [21]. The shear modulus of edible sunflower extracts was measured by a shear test. During the test, the tested edible sunflower extracts were naturally placed on the center of the square platform of a TA-XT plus texture analyzer, and a P/WB shear probe was used to compress the sample. The speed before compression was 1 mm/s, the speed during compression was 0.05 mm/s, and the speed after compression was 5 mm/s. The three test samples of edible sunflower seeds, edible sunflower plates (broken) and edible sunflower petioles were tested 10 times, and the average value was taken. The shear modulus of edible sunflower seeds was calculated by Formula (2) to be 56.3 MPa. The shear modulus of the discs (broken) was 35.5 MPa, and the shear modulus of the sunflower petioles was 53.1 MPa.

$$G = \frac{\tau}{\gamma} \tag{2}$$

where $G$ is the shear modulus, MPa; $\tau$ is the shear stress, N; and $\gamma$ is the shear strain.

### 2.3.3. Poisson's Ratio

When a material is stretched or compressed in one direction, the ratio of the absolute value of the transverse normal strain to the axial normal strain is called Poisson's ratio, which is an elastic constant that reflects the transverse deformation of the material [22]. Through the determination of the elastic modulus and shear modulus of edible sunflower, Poisson's ratio of edible sunflower was calculated using (3), and the calculation results are shown in Table 2.

$$\mu = \frac{E}{2G} - 1 \qquad (3)$$

**Table 2.** Poisson's ratio of edible sunflower.

| Type | Elastic Modulus (MPa) | Shear Modulus (MPa) | Poisson's Ratio |
|---|---|---|---|
| Sunflower seeds | 147.6 | 56.3 | 0.31 |
| Sunflower plates (broken) | 92.4 | 35.5 | 0.30 |
| Sunflower petioles | 133.4 | 53.1 | 0.26 |

In the formula, $\mu$ is Poisson's ratio; $E$ is the modulus of elasticity, MPa; and $G$ is the shear modulus, MPa.

### 2.4. Static Friction Coefficient

There is a large amount of relative movement in the extraction of sunflower seeds during the cleaning process, and the accurate measurement of the static friction coefficient can directly affect the simulation results of the cleaning process [23]. The static friction coefficient was measured using the inclined plane method [24], and a self-designed agricultural material static friction coefficient measuring device was used for the measurement. During the measurement, the sunflower extract was placed on the surface of the material to be tested, and the adjustment handle was slowly turned to increase the inclined plane angle. The movement state of the edible sunflower protruding object was observed. When a downward sliding trend on the material to be tested was observed, the number on the digital display of the angle measuring instrument was read and recorded as σ, and the difference value between the edible sunflower protruding object and the contact material was calculated. The coefficient of static friction f = tanσ, and each test was repeated 10 times, as shown in Figure 4. Table 3 shows the measurement results of the static friction coefficient of edible sunflower extracts.

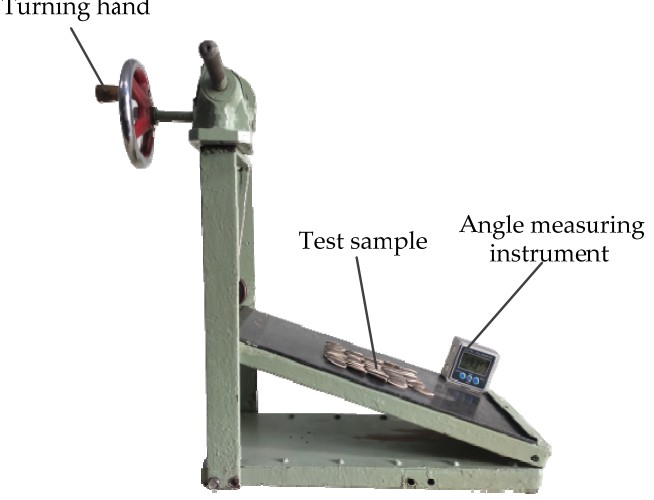

**Figure 4.** Test of the static friction coefficient of edible sunflower extrudate.

**Table 3.** The static friction coefficient and collision recovery coefficient of edible sunflower.

| Type | Coefficient of Static Friction | Collision Recovery Coefficient |
|---|---|---|
| Sunflower seeds-Sunflower seeds | 0.57 | 0.57 |
| Sunflower seeds-Sunflower plates (broken) | 0.55 | 0.67 |
| Sunflower seeds-Sunflower petioles | 0.54 | 0.53 |
| Sunflower plates (broken)-Sunflower plates (broken) | 0.81 | 0.56 |
| Sunflower plates (broken)-Sunflower petioles | 0.82 | 0.40 |
| Sunflower petioles-Sunflower petioles | 0.83 | 0.48 |
| Sunflower seeds-Rigid sieve plate | 0.73 | 0.58 |
| Sunflower plates (broken)-Rigid sieve plate | 0.61 | 0.50 |
| Sunflower petioles-Rigid sieve plate | 0.78 | 0.50 |

*2.5. Collision Recovery Coefficient*

The collision recovery coefficient e is a parameter that measures the ability of objects to recover from deformation after collision. It is the ratio of the normal velocity $V_1'$ of two objects after collision and the normal relative velocity $V_1$ before collision [25]. The free-fall test was used to determine the impact recovery coefficient of edible sunflower extracts [26]. The measurement principle diagram is shown in Figure 5. The height $H$ of the measured material falling after colliding with the 45° collision plate was set, and the resulting horizontal distance was $L$. The material was measured during free fall from height $H$, and the time to reach the collision plate was $t_1$. After collision with the collision plate, the movement of the material was parabolic. The speed of the free fall before collision was $V_2$, and the horizontal speed after collision was $V_x$. The time to reach the glued board was $t_2$.

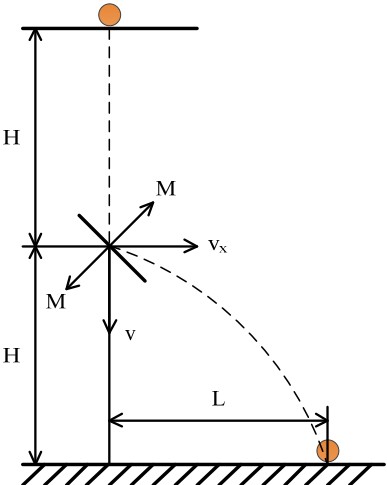

**Figure 5.** Schematic diagram of the measurement of the collision recovery coefficient.

According to the principle of kinematics:

$$\begin{cases} v = \sqrt{2gH} \\ H = \dfrac{gt^2}{2} \\ t_1 = t_2 = \sqrt{\dfrac{2H}{g}} \end{cases} \qquad (4)$$

$$V_X = \frac{L}{t_2} = \frac{L}{\sqrt{\dfrac{2H}{g}}} \qquad (5)$$

The subvelocity of the edible sunflower protruding object in the normal direction before and after the collision point is

$$\begin{cases} v_1 = v_x \sin 45° \\ v_1' = v \sin 45° \end{cases} \tag{6}$$

Therefore, the collision recovery coefficient of the edible sunflower protruding object is

$$e = \frac{v_1'}{v_1} = \frac{v_x \sin 45°}{v \sin 45°} = \frac{L}{2H} \tag{7}$$

A self-designed drop collision device and a high-speed camera (i-SPEED3) were used in the test. During the test, the edible sunflower protruding material fell from a height of 20 cm, bounced after colliding with the measurement sample plate, and finally landed on the bottom plate. The high-speed camera recorded the entire rebound process of the edible sunflower protruding object. The test was repeated 10 times. The test results are shown in the Table 3.

### 2.6. Angle of Repose

The angle of repose of edible sunflower was measured using a homemade angle of repose tester. As shown in Figure 6, the funnel length was 130 mm, and the inner diameter of the feed opening and the blanking opening were 114 mm and 30 mm, respectively. Thirty grams of edible sunflower was tested and poured from the funnel inlet in a free fall manner. When all the leached objects were completely still on the bottom steel plate, the camera was used to take the image of the leached objects, and MATLAB was used to ash the image of the edible sunflower leached objects. Binarization processing and boundary pixel extraction were conducted. The pixel coordinates were input into Origin software to fit the boundary line. The experiment was repeated 10 times, the average value of the accumulation angle of the edible sunflower protuberance was 24.68°, and the angle of repose boundary was fitted as shown in Figure 6.

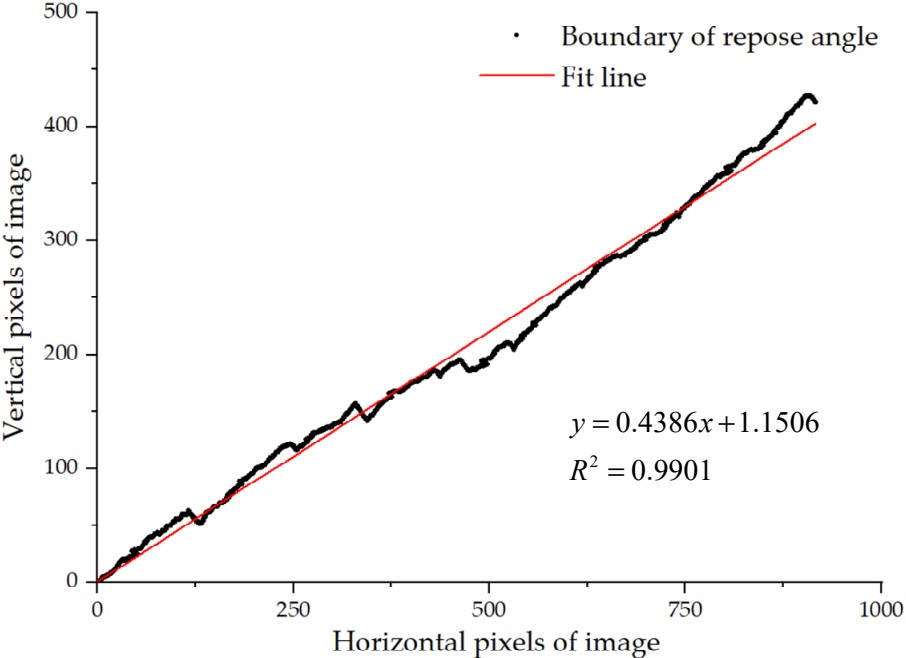

**Figure 6.** Fitting diagram of the unilateral angle of the repose boundary of edible sunflower extrudate.

## 3. Establishment of a Discrete Element Model of Edible Sunflower Extraction and Calibration of Simulation Parameters

### 3.1. Establishment of a Discrete Element Simulation Model

Through the early determination of the physical characteristics of edible sunflower extracts, the three-dimensional modeling software SolidWorks was used to model the actual size of the edible sunflower extracts, and then the extracted object model was converted into an IGES model and imported into the discrete element simulation software EDEM. The single spherical particles in the EDEM software filled the edible sunflower exudate model to establish a simulation model of the edible sunflower exudate [27,28], as shown in Figure 7. In the EDEM simulation test, the measurement model of the simulation test of the angle of repose of the edible sunflower exudates was modeled according to the actual size of the homemade angle of the repose tester, as shown in Figure 7. At the beginning of the simulation, edible sunflower exudates were dynamically generated from the feed opening above the funnel in a free-falling motion. After 4 s of simulation time, all edible sunflower exudates fell on the bottom plate and remained stationary, forming an angle of repose for the accumulation of edible sunflower exudates.

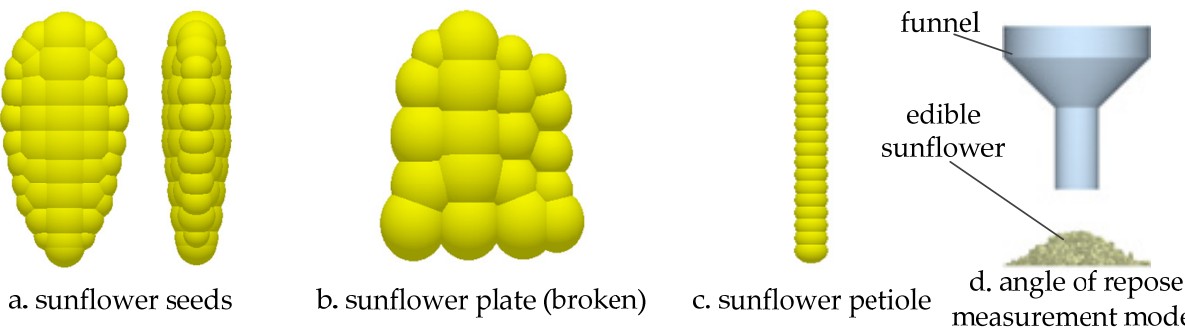

a. sunflower seeds　　　b. sunflower plate (broken)　　　c. sunflower petiole　　　d. angle of repose measurement model

**Figure 7.** Simulation model of edible sunflower exudates.

### 3.2. Calibration of Discrete Element Simulation Parameters

Determination of Significant Influence Parameters

The Plackett–Burman mode of Design-Expert 11 software was used to design the experiment. The test parameters were selected according to the results of the material test, and the angle of repose after accumulation of edible sunflower was used as the response value to screen out the parameters that have a significant impact on the response value [29]. The Plackett–Burman test program has a total of 24 test parameters, which are represented by X1-X24, and three virtual parameters are set, which are represented by X25-X27. The high (+1) and low (−1) levels of each parameter are the measured maximum and minimum values, respectively, as shown in Table 4. After each set of simulation experiments, the method of obtaining the angle of repose of the simulation model of the sunflower exudate is the same as the method of physically measuring the angle of repose. The Plackett–Burman test scheme and results are shown in Table 5.

**Table 4.** Plackett–Burman test parameter range table.

|  | Test Parameters | Low Level | High Level |
|---|---|---|---|
| Poisson's ratio | Sunflower seeds X1 | 0.21 | 0.41 |
|  | Sunflower plates (broken) X2 | 0.2 | 0.4 |
|  | Sunflower petioles X3 | 0.16 | 0.36 |
| Shear modulus (MPa) | Sunflower seeds X4 | 40.1 | 70.5 |
|  | Sunflower plates (broken) X5 | 23.4 | 49.3 |
|  | Sunflower petioles X6 | 33.9 | 63 |

**Table 4.** *Cont.*

| | Test Parameters | Low Level | High Level |
|---|---|---|---|
| | Sunflower seeds-Sunflower seeds X7 | 0.47 | 0.72 |
| | Sunflower seeds-Sunflower plates (broken) X8 | 0.52 | 0.58 |
| | Sunflower seeds-Sunflower petioles X9 | 0.5 | 0.65 |
| Coefficient of static friction | Sunflower plates (broken)-Sunflower plates (broken) X10 | 0.77 | 0.86 |
| | Sunflower plates (broken)-Sunflower petioles X11 | 0.79 | 0.86 |
| | Sunflower petioles-Sunflower petioles X12 | 0.81 | 0.85 |
| | Sunflower seeds-Rigid sieve plate X13 | 0.69 | 0.76 |
| | Sunflower plates (broken)-Rigid sieve plate X14 | 0.59 | 0.65 |
| | Sunflower petioles-Rigid sieve plate X15 | 0.74 | 0.82 |
| | Sunflower seeds-Sunflower seeds X16 | 0.5 | 0.68 |
| | Sunflower seeds-Sunflower plates (broken) X17 | 0.63 | 0.73 |
| | Sunflower seeds-Sunflower petioles X18 | 0.5 | 0.58 |
| | Sunflower plates (broken)-Sunflower plates (broken) X19 | 0.48 | 0.6 |
| Collision recovery coefficient | Sunflower plates (broken)-Sunflower petioles X20 | 0.33 | 0.48 |
| | Sunflower petioles-Sunflower petioles X21 | 0.4 | 0.55 |
| | Sunflower seeds-Rigid sieve plate X22 | 0.48 | 0.7 |
| | Sunflower plates (broken)-Rigid sieve plate X23 | 0.43 | 0.58 |
| | Sunflower petioles-Rigid sieve plate X24 | 0.43 | 0.55 |

**Table 5.** Plackett–Burman test results.

| Number | Angle of Repose (°) | Number | Angle of Repose (°) |
|---|---|---|---|
| 1 | 24.88 | 15 | 30.74 |
| 2 | 24.18 | 16 | 26.74 |
| 3 | 27.64 | 17 | 27.53 |
| 4 | 33.35 | 18 | 24.84 |
| 5 | 30.96 | 19 | 32.94 |
| 6 | 27.27 | 20 | 26.83 |
| 7 | 30.73 | 21 | 30.51 |
| 8 | 27.14 | 22 | 27.45 |
| 9 | 27.80 | 23 | 29.52 |
| 10 | 29.47 | 24 | 26.65 |
| 11 | 26.73 | 25 | 33.69 |
| 12 | 27.56 | 26 | 30.68 |
| 13 | 29.46 | 27 | 33.53 |
| 14 | 30.89 | 28 | 30.79 |

Design-Expert 11 software was used to analyze the variance of the test results and obtain the significance results of each simulation parameter, as shown in Table 6. From Table 6, it can be seen that $p < 0.01$ for the shear modulus of sunflower grain-sunflower grain, the static friction coefficient of sunflower grain-sunflower grain, and the coefficient of recovery of sunflower grain-sunflower grain, which have extremely significant influence on the angle of repose in the simulation experiment. For other simulation test parameters, $p > 0.05$, which indicates minimal influence on the angle of repose of the simulation test.

### 3.3. The Steepest Climbing Test Design

The steepest climbing test is carried out on the three significant parameters of sunflower kernel-sunflower kernel shear modulus, static friction coefficient and collision recovery coefficient obtained from the Plackett–Burman test, which can quickly determine the optimal selection of the significant factors. For the value range, the relative error between the simulated angle of repose and the actual angle of repose is used as the evaluation index. The results are shown in Table 7. With the increase in the value of each saliency parameter, the relative error first decreases and then increases. For No. 3, the relative error

is the smallest, so it can be determined that the level range of the optimal parameter is approximately No. 3 [30].

**Table 6.** Significance analysis of Plackett–Burman test parameters.

| Parameter | Degree of Freedom | Sum of Squares | F Value | *p* Value |
|---|---|---|---|---|
| X1 | 1 | 1.80 | 3.95 | 0.1411 |
| X2 | 1 | 0.3214 | 0.7051 | 0.4627 |
| X3 | 1 | 0.2721 | 0.5968 | 0.4961 |
| X4 | 1 | 56.52 | 123.97 | 0.0016 ** |
| X5 | 1 | 0.0001 | 0.0001 | 0.9918 |
| X6 | 1 | 0.3214 | 0.7051 | 0.4627 |
| X7 | 1 | 63.12 | 138.46 | 0.0013 ** |
| X8 | 1 | 0.7957 | 1.75 | 0.2782 |
| X9 | 1 | 3.98 | 8.74 | 0.0597 |
| X10 | 1 | 3.28 | 7.19 | 0.0750 |
| X11 | 1 | 0.7426 | 1.63 | 0.2917 |
| X12 | 1 | 0.3432 | 0.7529 | 0.4494 |
| X13 | 1 | 0.0146 | 0.0321 | 0.8692 |
| X14 | 1 | 2.02 | 4.43 | 0.1260 |
| X15 | 1 | 1.57 | 3.45 | 0.1601 |
| X16 | 1 | 41.29 | 90.56 | 0.0025 ** |
| X17 | 1 | 4.35 | 9.55 | 0.0537 |
| X18 | 1 | 2.04 | 4.48 | 0.1247 |
| X19 | 1 | 0.1317 | 0.2888 | 0.6283 |
| X20 | 1 | 0.4784 | 1.05 | 0.3810 |
| X21 | 1 | 1.71 | 3.75 | 0.1482 |
| X22 | 1 | 3.76 | 8.25 | 0.0640 |
| X23 | 1 | 0.1889 | 0.4144 | 0.5656 |
| X24 | 1 | 3.29 | 7.22 | 0.0746 |

Note: ** indicates that the impact is extremely significant ($p < 0.01$).

**Table 7.** The design scheme and results of the steepest climbing test.

| Number | Shear Modulus (MPa) | Coefficient of Static Friction | Collision Recovery Coefficient | Angle of Repose (°) | Relative Error (%) |
|---|---|---|---|---|---|
| 1 | 40.1 | 0.47 | 0.5 | 18.95 | 19.97 |
| 2 | 46.18 | 0.52 | 0.536 | 22.63 | 4.43 |
| 3 | 52.26 | 0.57 | 0.572 | 24.05 | 1.56 |
| 4 | 58.34 | 0.62 | 0.608 | 25.33 | 6.97 |
| 5 | 64.42 | 0.67 | 0.644 | 26.77 | 13.05 |
| 6 | 70.5 | 0.72 | 0.68 | 30.89 | 30.45 |

In the simulation test, the other nonsignificant parameters are the average values determined by the physical test. The Poisson ratio of edible sunflower seeds is 0.31, the Poisson ratio of edible sunflower disks (broken) is 0.3, the Poisson ratio of edible sunflower petioles is 0.26, the shear modulus of sunflower plates (broken) is 35.5, the shear modulus of sunflower petioles is 53.1, the static friction coefficient of sunflower grain-sunflower plates (broken) is 0.55, and the collision recovery coefficient is 0.67. The static friction coefficient of sunflower petioles is 0.54, the collision recovery coefficient is 0.53, the static friction coefficient of sunflower plates (broken)-sunflower plates (broken) is 0.81, the collision recovery coefficient is 0.56, the static friction coefficient of sunflower plates (broken)-sunflower plates (broken) is 0.82, the collision recovery coefficient is 0.4, the static friction coefficient of sunflower petioles-sunflower petioles is 0.83, the collision recovery coefficient is 0.48, the static friction coefficient of sunflower grain-rigid sieve plate is 0.73, and the collision recovery coefficient is 0.58. The static friction coefficient of discs (broken)-rigid sieve plate is 0.61, the collision recovery coefficient is 0.5, the static friction coefficient of sunflower petioles-rigid sieve plate is 0.78, and the collision recovery coefficient is 0.5.

### 3.3.1. Box–Behnken Experimental Design

Using the Box–Behnken mode of Design-Expert 11 software for testing, the test factors are 3, of which No. 2 and No. 4 are low ($-1$) and high ($+1$) levels, and No. 3 is the center point (0). The designed scheme and results are shown in Table 8. In the simulation test, other nonsignificant parameters are the average values determined by the physical test.

**Table 8.** Box–Behnken experimental design scheme and results.

| Number | Shear Modulus of Sunflower Seeds-Sunflower Seeds (MPa) | Coefficient of Static Friction of Sunflower Seeds-Sunflower Seeds | Collision Recovery Coefficient of Sunflower Seeds-Sunflower Seeds | Angle of Repose (°) |
|---|---|---|---|---|
| 1 | $-1$ | 0 | $-1$ | 21.48 |
| 2 | $-1$ | 0 | 1 | 22.39 |
| 3 | 1 | 0 | $-1$ | 23.95 |
| 4 | 1 | 0 | 1 | 26.48 |
| 5 | 0 | $-1$ | $-1$ | 21.34 |
| 6 | 0 | $-1$ | 1 | 22.49 |
| 7 | 0 | 1 | $-1$ | 25.22 |
| 8 | 0 | 1 | 1 | 26.53 |
| 9 | $-1$ | $-1$ | 0 | 20.77 |
| 10 | 1 | $-1$ | 0 | 21.63 |
| 11 | $-1$ | 1 | 0 | 22.15 |
| 12 | 1 | 1 | 0 | 26.67 |
| 13 | 0 | 0 | 0 | 24.15 |
| 14 | 0 | 0 | 0 | 24.72 |
| 15 | 0 | 0 | 0 | 24.65 |
| 16 | 0 | 0 | 0 | 23.96 |
| 17 | 0 | 0 | 0 | 24.37 |

Design-Expert 11 software was used to perform multiple regression analysis on the significant parameters of the Box–Behnken test and obtain the second-order regression model of the angle of repose and the significant parameters:

$$\begin{aligned} \theta = \ & 18.31 + 0.14X1 + 149.7X2 - 219.08X3 \\ & + 3.01X1X2 + 1.85X1X3 + 22.22X2X3 \\ & - 0.026X1^2 - 249X2^2 + 113.81X3^2 \end{aligned} \qquad (8)$$

Table 9 shows that X1, X2, X3, X1X2, and $X1^2$ have extremely significant effects on the angle of repose; $X2^2$ has significant effects on the angle of repose; X1X3, X2X3, and $X3^2$ have no significant effects on the angle of repose. $p < 0.0001$ is observed for the fitting model of the angle of repose, $p = 0.2805 > 0.05$ is observed for the lack-of-fit term, and the coefficient of variation CV = 1.59%, indicating that the equation is well fitted. The regression determination coefficient $R^2 = 0.983$ and the adjustment determination coefficient Adjust $R^2 = 0.9612$ are all close to 1, indicating that the angle of repose model is extremely significant and can truly reflect the situation, thereby predicting the target angle of repose.

### 3.3.2. Simulation Parameter Calibration and Test Verification

Taking the actual physical angle of repose as the target value, module optimization is used in Design-Expert 11 software to optimize the second-order regression equation so that the simulated angle of repose value is closest to the physical test angle of repose value, and a set of data similar to the physical test data is obtained. The optimal parameter combination of sunflower kernel-sunflower kernel shear modulus, static friction coefficient and collision recovery coefficient is 58.34 MPa, 0.62 and 0.608, respectively, and the values of other nonsignificant parameters are the same as those of the steepest climbing test.

**Table 9.** Box–Behnken test regression model analysis of variance.

| Source of Variance | Mean Square | Degree of Freedom | Sum of Square | *p* Value |
|---|---|---|---|---|
| model | 6.39 | 9 | 57.55 | <0.0001 ** |
| X1 | 17.82 | 1 | 17.82 | <0.0001 ** |
| X2 | 25.70 | 1 | 25.70 | <0.0001 ** |
| X3 | 4.35 | 1 | 4.35 | 0.0009 ** |
| X1X2 | 3.35 | 1 | 3.35 | 0.0018 ** |
| X1X3 | 0.6561 | 1 | 0.6561 | 0.0685 |
| X2X3 | 0.0064 | 1 | 0.0064 | 0.8378 |
| $X1^2$ | 3.74 | 1 | 3.74 | 0.0013 ** |
| $X2^2$ | 1.63 | 1 | 1.63 | 0.0116 * |
| $X3^2$ | 0.0916 | 1 | 0.0916 | 0.4480 |
| Residual | 0.1418 | 7 | 0.9927 | |
| Lack of fit | 0.1918 | 3 | 0.5753 | 0.2805 |
| Pure error | 0.1043 | 4 | 0.4174 | |
| Sum | | 16 | 58.54 | |

Note: ** indicates that the impact is extremely significant ($p < 0.01$) and * indicates that the impact is significant ($p < 0.05$).

To verify the authenticity of the optimal parameter combination after calibration of the sunflower exudates, the parameters obtained above were input into EDEM 2020 software for simulation, and the angles of repose of the sunflower exudates were 25.96°, 25.31°, and 23.47°, respectively, on average. The value is 24.91°. The actual physical test and simulated angles of repose of the sample were tested by two-sample *T* test, and $p = 0.827 > 0.05$, indicating that there was no significant difference between the simulated angle of repose and the actual physical test angle of repose. The actual physical angle of repose is compared with the optimal parameters. The relative error of the average value of the simulated angle of repose under the combination is 0.923%, and the experimental comparison result is shown in Figure 8 and Table 10.

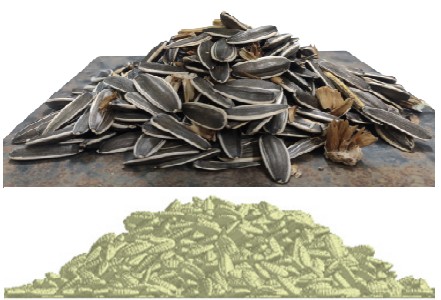

**Figure 8.** Comparison of the angle of repose test of edible sunflower extrudate.

**Table 10.** Comparison of the angle of repose test results of edible sunflower extrudate.

| NO. | The Simulated Angle of Repose (°) | The Actual Physical Angle of Repose (°) | Relative Error (%) |
|---|---|---|---|
| 1 | 25.96<br>25.31<br>23.47<br>24.91 | 24.68 | 0.923 |

## 4. Conclusions

The composition and basic physical properties of the extract from sunflower were obtained through physical tests using a homemade static friction coefficient measuring instrument. A collision device and a high-speed camera were used to obtain the static friction coefficient and the collision recovery coefficient between components of sunflower extract.

Taking the basic physical characteristics and mechanical characteristic parameters determined by the physical test as the basis of the simulation test, the Plackett–Burman test was carried out, and the parameters that have a significant effect on the simulation angle of repose were obtained: sunflower kernel-sunflower kernel shear modulus, static friction coefficient and collision recovery coefficient, to determine the value range of the significant parameters through the steepest climbing test.

On the basis of the steepest climbing test, the Box–Behnken test was carried out to obtain the second-order regression equation of the simulated angle of repose and the significant parameters, and the actual physical angle of repose (24.68°) was used as the target value to optimize the solution and obtain the best parameter combination.

Through the two-sample $T$ test, we obtained $p = 0.827 > 0.05$, indicating that there was no significant difference between the simulated angle of repose and the actual physical test angle of repose, and the relative error was 0.923%, which verifies the reliability of the optimal simulation parameter combination after calibration. The above verification test results show that the calibration results of simulation parameters of discrete element model of sunflower extract are true and reliable, which can provide important reference value for the simulation of mechanized operation of sunflower harvesting, cleaning, and other processes.

**Author Contributions:** Conceptualization, X.S. and B.L.; methodology, B.L.; software, X.S.; validation, X.S.; formal analysis, Y.L. and X.S.; data curation, X.S. and X.G.; writing—original draft preparation, X.S. and B.L.; writing—review and editing, B.L. and Y.L. All authors have read and agreed to the published version of the manuscript.

**Funding:** This research received no external funding.

**Institution Review Board Statement:** Not applicable.

**Informed Consent Statement:** Not applicable.

**Conflicts of Interest:** The authors declare no conflict of interest.

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
