# Peer review of "Parameter Measurement of Edible Sunflower Exudates and Calibration of Discrete Element Simulation Parameters"

_processes, doi:10.3390/pr10020185_

Round 1
Reviewer 1 Report
Processes
Parameter measurement of edible sunflower exudates and cali-2 bration of discrete element simulation parameters
The recent research focuses on conducting in-depth research on the calibration of discrete element model simulation parameters of edible sunflower extracts from the planting area of Urumqi County, Xinjiang. In this regard, the measured basic physical and mechanical parameters are used as the basis for the selection of discrete element simulation parameters, and the actual measured physical angle of repose is used as the target value.
This is a well-managed paper that investigated the related physical characteristics of edible sunflower extracts followed by the three-dimensional modelling software SolidWorks to model the actual size of the edible sunflower extracts. However, some points need to be addressed. Therefore, I am only recommending the approval of this paper with MINOR corrections as follows
- Please add the appropriate references for all the model equations presented in the manuscript
- Section 3.3.2: It might be possible to generate a specific table to compare the simulation results and the optimal ones.
- The possible targeted aspects related to this study might be suggested, at the end before the Conclusions, for further research
- Please use (Conclusions) instead of (Conclusion)
- Conclusions: it seems to me that you are writing an intensive Conclusions. Please shorten it by providing the most interesting findings
- Conclusions: please write precise and complete paragraphs without bullets
Author Response
请参阅附件。

Reviewer 2 Report
The paper: Parameter measurement of edible sunflower exudates and calibration of discrete element simulation parameters, by authors: Xiaoxiao Sun, Bin Li, Yang Liu and Xiaolong Gao present the experimental research of the parameters which will be used in discrete element method numerical simulations. The speciality of the used material (sunflower) was analysed. This investigation can be useful for the engineers involved in chemical and process industry, as well as for the researchers involved in DEM modelling. The paper obtained valuable results concerning sunflower parameters which can be used for future investigations. In my opinion, the paper could be published after a MAJOR revision. Comments: 1. The Abstract is well written. It is clear and covers all the major findings in the article. There is a mistake: sexual parameters should be texture parameters. 2. The Introduction section is well written. Please emphasize the novelty of your research. 3. The section Materials and Methods-Maybe Table 1 can be simplified. There is no need to repeat Length×Diameter×Width (mm×mm×mm), Moisture content (%) and Density (kg/m3 ) in 3 colums. 4. Figure 4. Test of the static friction coefficient of edible sunflower extrudate. It is not clear. I do not see the need for all these figures. One is enough. 5. Table 7. The design scheme and results of the steepest climbing test. You repeat sunflower seeds unnecessarily.Author Response
请参阅附件。

Round 2
Reviewer 2 Report
The authors changed the manuscript according to my remarks.In my opinion,the manuscript can be published in the present form.